# Insect size responses to climate change vary across elevations according to seasonal timing

César R. Nufio[1,2], Monica M. Sheffer[3,4], Julia M. Smith[3], Michael T. Troutman[5], Simran J. Bawa[4], Ebony D. Taylor[5], Sean D. Schoville[5], Caroline M. Williams[4], Lauren B. Buckley [3]*

1 Howard Hughes Medical Institute, Chevy Chase, Maryland, United Sates of America, 2 University of Colorado Natural History Museum, University of Colorado, Boulder, Colorado, United States of America, 3 Department of Biology, University of Washington, Seattle, Washington, United States of America, 4 Department of Integrative Biology, University of California, Berkeley, California, United States of America, 5 Department of Entomology, University of Wisconsin, Madison, Wisconsin, United States of America

* lbuckley@uw.edu

## Abstract

Body size declines are a common response to warming via both plasticity and evolution, but variable size responses have been observed for terrestrial ectotherms. We investigate how temperature-dependent development and growth rates in ectothermic organisms induce variation in size responses. Leveraging long-term data for six montane grasshopper species spanning 1,768–3 901 m, we detect size shifts since ~1960 that depend on elevation and species' seasonal timing. Size shifts have been concentrated at low elevations, with the early emerging species (those that overwinter as juveniles) increasing in size, while later season species are becoming smaller. Interannual temperature variation accounts for the size shifts. The earliest season species may be able to take advantage of warmer conditions accelerating growth during early spring development, whereas warm temperatures may adversely impact later season species via mechanisms such as increased rates of energy use or thermal stress. Grasshoppers tend to capitalize on warm conditions by both getting bigger and reaching adulthood earlier. Our analysis further reinforces the need to move beyond expectations of universal responses to climate change to consider how environmental exposure and sensitivity vary across elevations and life histories.

## Introduction

Declining body size is a commonly anticipated response to anthropogenic climate warming [1,2], but variation in responses between taxa and environments challenges this generality [3,4]. Body size impacts surface area to volume ratios, fluxes of energy and matter between organisms and their environment, reproductive output and fitness, and the outcome of competitive and exploitative interactions; making it a key trait to link organismal responses to climate change to their ecosystem consequences [5–10]. Intrinsic (genetic and plastic differences) and extrinsic (environmental) factors contribute to body size variation, and

**Data availability statement:** Data and code are available in Dryad (https://doi.org/10.5061/dryad.wwpzgmst6) and GitHub (https://github.com/lbuckley/HopperBodysize).

**Funding:** This work was supported by the National Science Foundation (DEB-1951356 to L.B.B., DEB-1951588 to S.D.S. and DEB-1951364 to C.M.W.). No funder played any role in study design, data collection or analysis, decision to publish, or preparation of the manuscript.

**Competing interests:** The authors have declared that no competing interests exist.

**Abbreviations :** ANOVA, analysis of variance; doy, day of year; LMEs, linear mixed-effects; LTER, Long-Term Ecological Research program; TSR, temperature-size rule.

disentangling these factors is necessary to making general predictions of biological responses to climate change [11–13].

Warmer environmental temperatures are associated with smaller body size on a range of spatial and temporal scales. The temperature-size rule (TSR) addresses phenotypic plasticity in ectotherms, wherein >80% of ectothermic organisms mature faster and at smaller adult sizes in warm compared to cool conditions ("hotter is smaller" [14–16], (Fig 1). Organisms tend to be smaller in warmer geographic locations (e.g., low latitude and elevation) both within (James' Rule) and across species (Bergmann's Rule). Both rules encompass evolutionary drivers. They were originally formulated for endotherms and have mixed occurrence for ectotherms [9]. Terrestrial ectotherms show the greatest variability in size responses to warming, compared to aquatic or endothermic species, due to their small body size and differences in resource and oxygen limitation and allocation [7,17–19]. Small-bodied, terrestrial organisms have a high surface-area-to-volume ratio, high mass-specific metabolic rates, and reduced thermal inertia, making them more sensitive to environmental shifts, but with lower peak body temperature during thermal extremes [1,6,18].

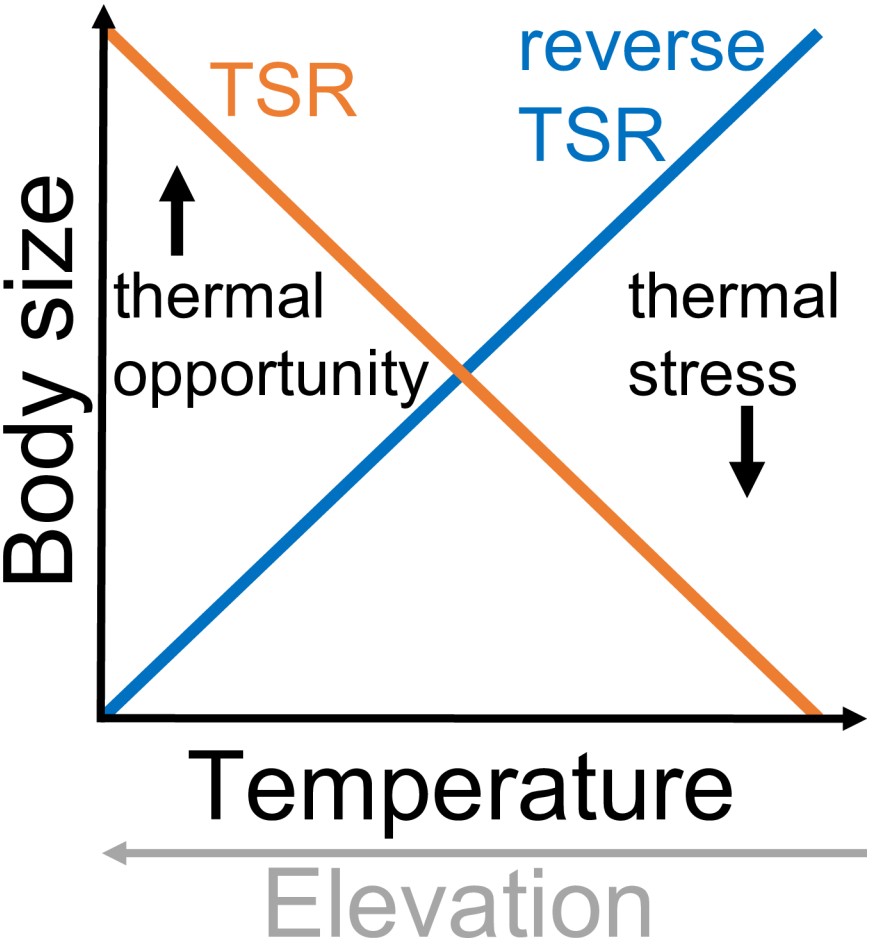

**Fig 1. Hypotheses for grasshopper body size response to temperature gradients and change.** Most ectotherms exhibit "hotter is smaller" temperature responses and elevation clines (temperature-size rule [TSR]), but season-limited ectotherms with few generations often exhibit reverse TSR. Warming may additionally increase body size at low temperatures/high elevations via increased thermal opportunity. Conversely, thermal stress induced by warming at high temperatures/low elevations may decrease body size.

Many terrestrial arthropods show "hotter is bigger" latitudinal or elevational body size clines, but a number of species show reverse or sawtooth clines [20]. Selection may favor large body size at the expense of long generation time when seasons are short relative to generation time, but favor rapid generations when there are many generations possible per year [21]. This suggests that life history, and in particular seasonal timing, is likely to determine body size responses to climate change [4].

In insects, natural history collections have illustrated body size increases, decreases, and stasis over the past 50–200 years of climate change [22–27]. Detecting morphological shifts in response to climate change is hampered by many natural history collections being composed of specimens with limited spatial and temporal replication and potential collection biases [28,29]. Here, we use an exceptional natural history collection to measure body size changes in a grasshopper community along an elevational gradient in the Colorado Rocky Mountains. The collection, initiated by Gordon Alexander in 1958, allows us to assess responses to more than 50 years of anthropogenic climate change. The collection consists of historic (primarily 1958–1960) and modern (2006–2015) samples, and survey records detailing phenology [30]. Grasshoppers are an important example of an ectotherm group that reverses the TSR ("hotter is bigger") [7,31]. The physiological processes related to feeding have evolved to be optimal under warm temperatures in many grasshopper species, which may lead to greater increases in growth than development at warm temperatures [32]. Additionally, many species have a single annual generation, which prohibits fitness increases associated with additional generations, and thus may select for large body size [33,34].

We predict that warmer conditions will increase body size, consistent with observed "hotter is bigger" elevation clines, although species may differ in the strength of responses [35] (Fig 1). A greater degree of warming and higher developmental plasticity may augment body size increases at high elevations [36–40]. High elevation populations and early season species may benefit from increased thermal opportunities in the spring and early summer months, while low elevation populations and late season species may be negatively impacted by hot temperatures during mid-summer, truncating the end of the growing season [17] (Fig 1).

## Results

### Clinal patterns and shifts through time

Growing season temperatures were used as a metric to capture the mean daily temperatures during the period of grasshopper development and activity (March through August, day of year [doy] 60–243). Growing season temperatures at our study sites decreased with increasing elevation and exhibited pronounced interannual variability (Fig 2). Growing season temperatures have warmed considerably over recent decades, with greater warming at higher elevations.

We quantified grasshopper body size through measurements of femur length in modern and historic specimens. We assess determinants of grasshopper body sizes using linear mixed-effects (LMEs) models and analysis of variance (ANOVA). We focus on whether size responses vary with elevation and species' seasonal timing and account for year and species as random effects. Our six focal species are well sampled across years, have a single annual generation, and span a range of seasonal timing ($n = 3,413$ specimens). *Eritettix simplex* and *Xanthippus corallipes* overwinter as juveniles (nymphs), and thus reach adulthood as early as May. The remaining species, which overwinter as eggs, reach adulthood in approximately mid-June (early season: *Aeropedellus clavatus*, *Melanoplus boulderensis*) or late July (late season: *Camnula pellucida*, *Melanoplus sanguinipes*) [36].

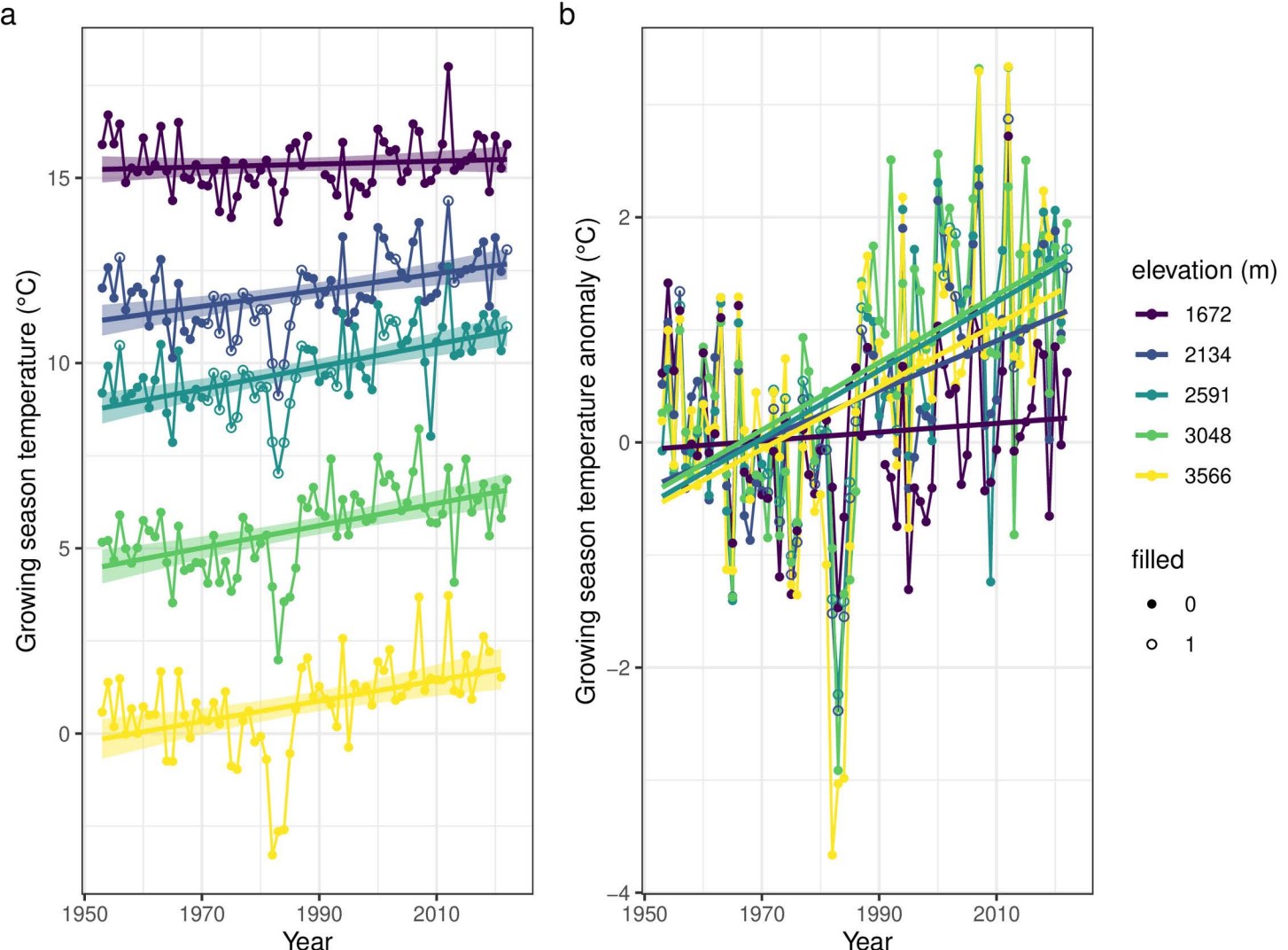

**Fig 2. Growing season temperatures have increased over recent decades, particularly at higher elevations. (a)** Growing season (March through August) means of daily mean temperature have increased over time (linear regressions trends ± SE shown). **(b)** Depicting the temperatures as anomalies highlights the greater warming at higher elevations (year: $F_{[1,342]} = 82.92$, $P < 10^{-16}$; elevation: $F_{[1,342]} = 6.06$, $P = 0.01$; year * elevation: $F_{[1,342]} = 10.19$, $P < 0.01$). Hollow dots indicate years where data from other sites were used to fill missing data. The data and code needed to generate this figure can be found in Dryad (https://doi.org/10.5061/dryad.wwpzgmst6).

Grasshopper body size declines as elevation increases (the predicted "hotter is bigger" pattern), with shallower declines for the late season species (Fig 3 and Fig A in S1 Text). Females are substantially larger than males in all species. An ANOVA for historic (pre-2000) body sizes detects significant main effects of elevation ($F_{[1,791]} = 52.96$, $P < 10^{-12}$) and sex ($F_{[1,791]} = 1380.39$, $P < 10^{-15}$) and significant interactions including a three-way interaction between elevation, sex, and seasonal timing ($F_{[1,791]} = 10.95$, $P < 10^{-4}$). Analogous significant effects are observed for current body sizes (Fig 3 and Fig A in S1 Text).

We probed how climate change is altering the size clines by calculating anomalies from average body sizes for each species, elevation, and sex over a baseline period of 1950–1980, which is commonly used in climate studies. Elevation and species' seasonal timing significantly interact with time period to influence body size changes (Table 1). Size responses vary considerably

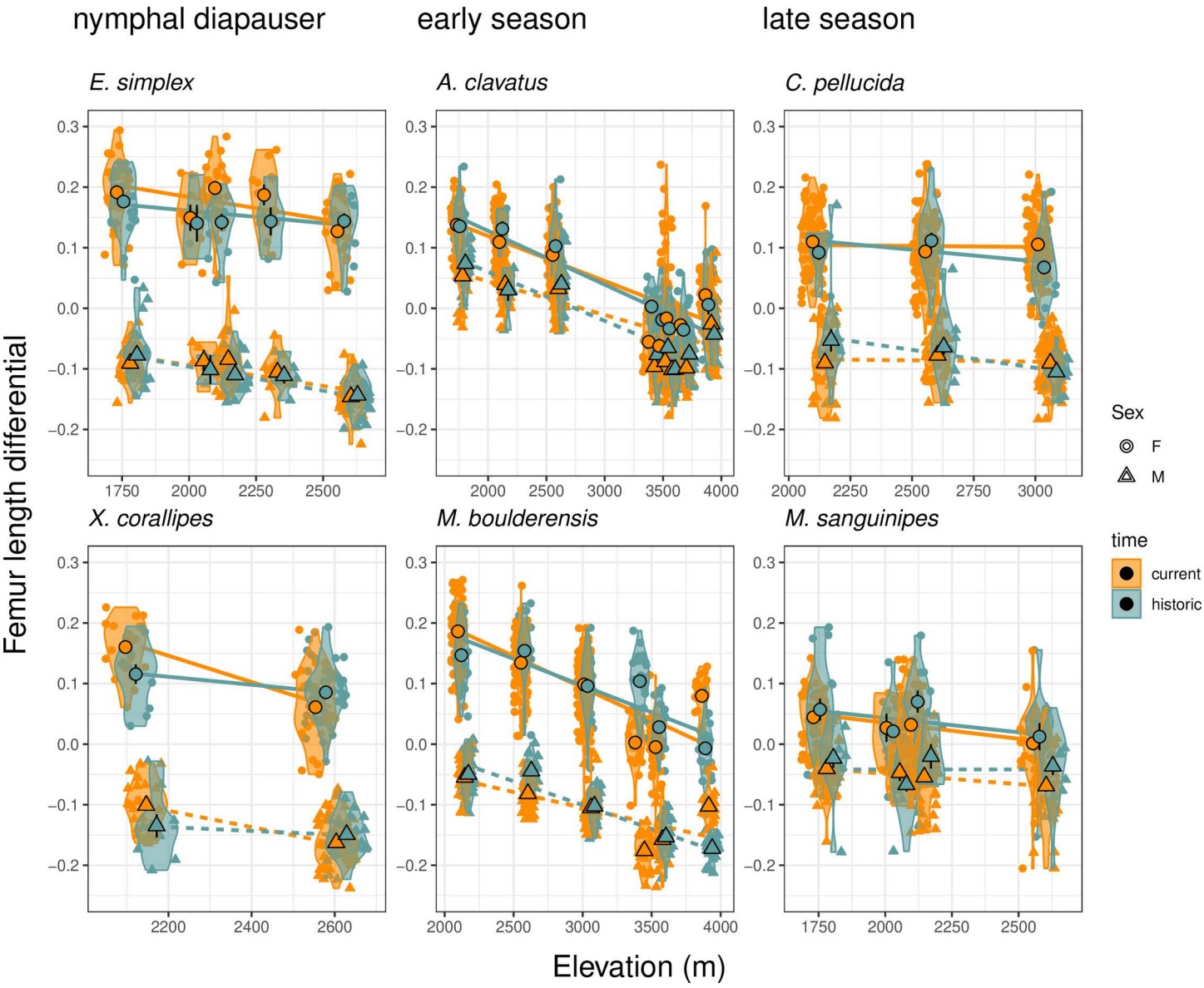

**Fig 3. Body size (femur length) declines with elevation, females are substantially larger than males, and the body size clines vary with seasonal timing.** We depict both violin plots and population means ± SE by sex and elevation. Data are proportional deviations from each species' mean size across the historic period. The species are arranged by seasonal timing from the earliest season, nymphal diapausing species to late season species (left to right then top to bottom). Body size changes between the historic and current time period differ based on seasonal timing and elevation. The slope of the relationship with elevation during the historic period (accounting for sex as an intercept) generally declines with later seasonal timing (slope + SE = *Eritettix simplex*: −0.03 ± 0.01, *Xanthippus corallipes*: −0.09 ± 0.03, *Aeropedellus clavatus*: −0.05 ± 0.00, *Melanoplus boulderensis*: −0.06 ± 0.01, *Camnula pellucida*: −0.01 ± 0.01, *Melanoplus sanguinipes*: −0.03 ± 0.01). The data and code needed to generate this figure can be found in Dryad (https://doi.org/10.5061/dryad.wwpzgmst6).

across species and elevations, but nymphal diapausing species have tended to increase size at lower elevations; early season species have shown little change or elevational trend; and late season species have tended to decrease size at low elevation since the historic period (Fig 4).

## Developmental responses to temperature

Do these body size changes represent temperature responses, consistent with shifts in growth and development rates? To address this, we compare the model with time period as a

**Table 1. ANOVA results examining how the body size anomaly varies over time (historic or current), with the growing season temperature anomaly (Tgs), and with time and the Tgs.** We additionally account for elevation (elev) and seasonal timing (SpTiming: nymphal diapauser, early season, or late season). We report model Akaike and Bayesian information criterion (AIC and BIC, respectively), where lower values indicate better model performance. We also report numerator and denominator degrees of freedom (NumDF and DenDF, respectively), the $F$-value statistics, and the $P$-value and its significance (*$p < 0.05$, **$p < 0.01$, ***$p < 0.001$).

| | | Time<br>AIC = 5,483, BIC = 5,568 | | | | Temperature<br>AIC = 5,423, BIC = 5,508 | | | | Time + temperature<br>AIC = 5,426, BIC = 5,583 | | |
|---|---|---|---|---|---|---|---|---|---|---|---|---|
| | NumDF | DenDF | $F$-value | Pr(>$F$) | | DenDF | $F$-value | Pr(>$F$) | | DenDF | $F$-value | Pr(>$F$) |
| Tgs | 1 | | | | | 287.68 | 3.21 | 0.07 | | 1543.29 | 0.05 | 0.82 |
| time | 1 | 60.24 | 0.05 | 0.82 | | | | | | 1074.7 | 0 | 0.99 |
| elev | 1 | 1134.71 | 0.54 | 0.46 | | 1913.48 | 13.76 | 0 | *** | 1083.1 | 0.96 | 0.33 |
| SpTiming | 2 | 55.78 | 0.19 | 0.83 | | 55.06 | 1.48 | 0.24 | | 921.31 | 0.24 | 0.78 |
| Tgs:time | 1 | | | | | | | | | 1543.29 | 0 | 0.98 |
| Tgs:elev | 1 | | | | | 2168.9 | 12.71 | 0 | *** | 1127.21 | 1.16 | 0.28 |
| time:elev | 1 | 1134.71 | 0.37 | 0.54 | | | | | | 1083.1 | 0.01 | 0.94 |
| Tgs:SpTiming | 2 | | | | | 268.6 | 6.09 | 0 | ** | 1541.54 | 0.04 | 0.96 |
| time:SpTiming | 2 | 55.78 | 0.85 | 0.43 | | | | | | 921.31 | 0.64 | 0.53 |
| elev:SpTiming | 2 | 1238.69 | 7.99 | 0 | *** | 2014.82 | 28.16 | 0 | *** | 1051.13 | 1.69 | 0.18 |
| Tgs:time:elev | 1 | | | | | | | | | 1127.21 | 0 | 0.96 |
| Tgs:time:SpTiming | 2 | | | | | | | | | 1541.54 | 1.25 | 0.29 |
| Tgs:elev:SpTiming | 2 | | | | | 2180.28 | 13.52 | 0 | *** | 1056.96 | 0.7 | 0.5 |
| time:elev:SpTiming | 2 | 1238.69 | 4.14 | 0.02 | * | | | | | 1051.13 | 0.28 | 0.76 |
| Tgs:time:elev:SpTiming | 2 | | | | | | | | | 1056.96 | 0.1 | 0.91 |

predictor of body size anomalies to models incorporating interannual variation in a series of different temperature measurements, including growing season temperature, prior summer temperature, and developmental temperature. ANOVAs indicate that temperature changes can account for size differences across time periods (Table 1). Warm growing season temperatures tend to increase the size of nymphal diapausing species at lower elevations (Fig 5 and Table 1). Early season species exhibit mixed responses to growing season temperatures. Late season species are smaller when spring temperatures are warmer, converse to the response of nymphal diapausing species. The interactions of time period or temperature with elevation and seasonal timing are no longer significant when including both time period and temperature in the model, and model performance declines, suggesting that the change in body size across time periods is largely attributable to interannual variation in temperature rather than any evolutionary shifts between time periods (Table 1). We also tested for different responses to interannual temperatures between modern and historic time periods (statistically, an interaction between temperature and time period). The lack of a significant interaction between time period and temperature (Table 1) suggests that plasticity is acting similarly across the time periods.

We next examined temperatures of the previous summer, which are expected to influence the initial development of nymphal diapausing species and the other species via maternal effects and rates of egg development and energy use. Summer temperatures have increased most strongly at high elevations (Fig B in S1 Text). We find that body size responses to temperature are similar during the growing season and the previous summer (Fig C in S1 Text). A four-way interaction suggests that grasshoppers at different elevations and with different seasonal timing are responding differently to temperature now than they did historically, indicating the potential for evolutionary responses between time periods (Table A in S1 Text).

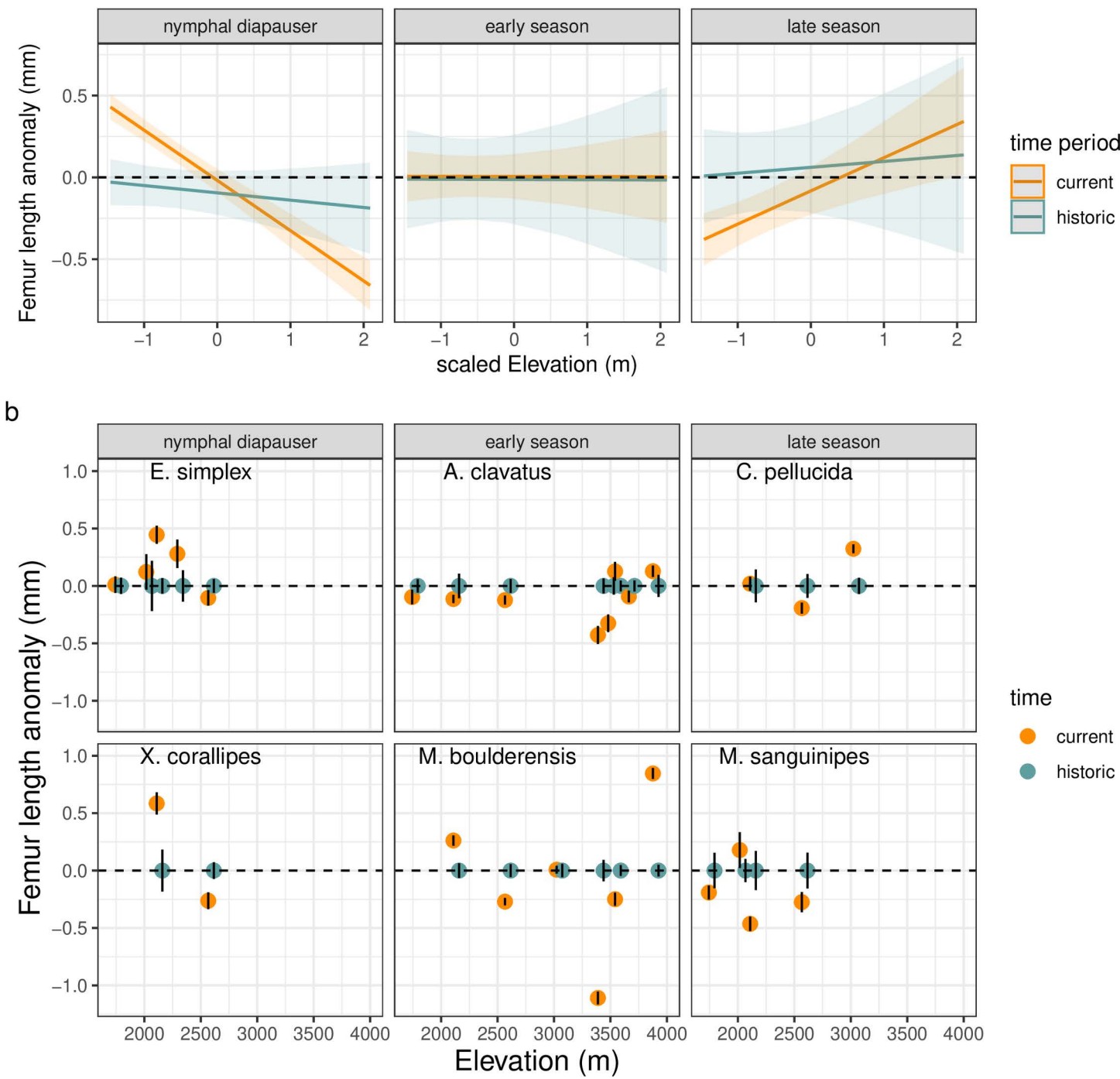

**Fig 4. Seasonal timing and elevation interact as determinants of changes in femur length from historic means (accounting for species, elevation, and sex).** Nymphal diapausing species have tended to increase size at lower elevations, early season species have shown little change or elevational trend, and late season species have tended the decrease size at low elevation since the historic period. The trends are depicted as **(a)** model estimates and standard errors (SEs) and **(b)** data means ± SE per year and site for species arranged by seasonal timing from the earliest season, nymphal diapausing species to late season species (left to right and top to bottom). The data and code needed to generate this figure can be found in Dryad (https://doi.org/10.5061/dryad.wwpzgmst6).

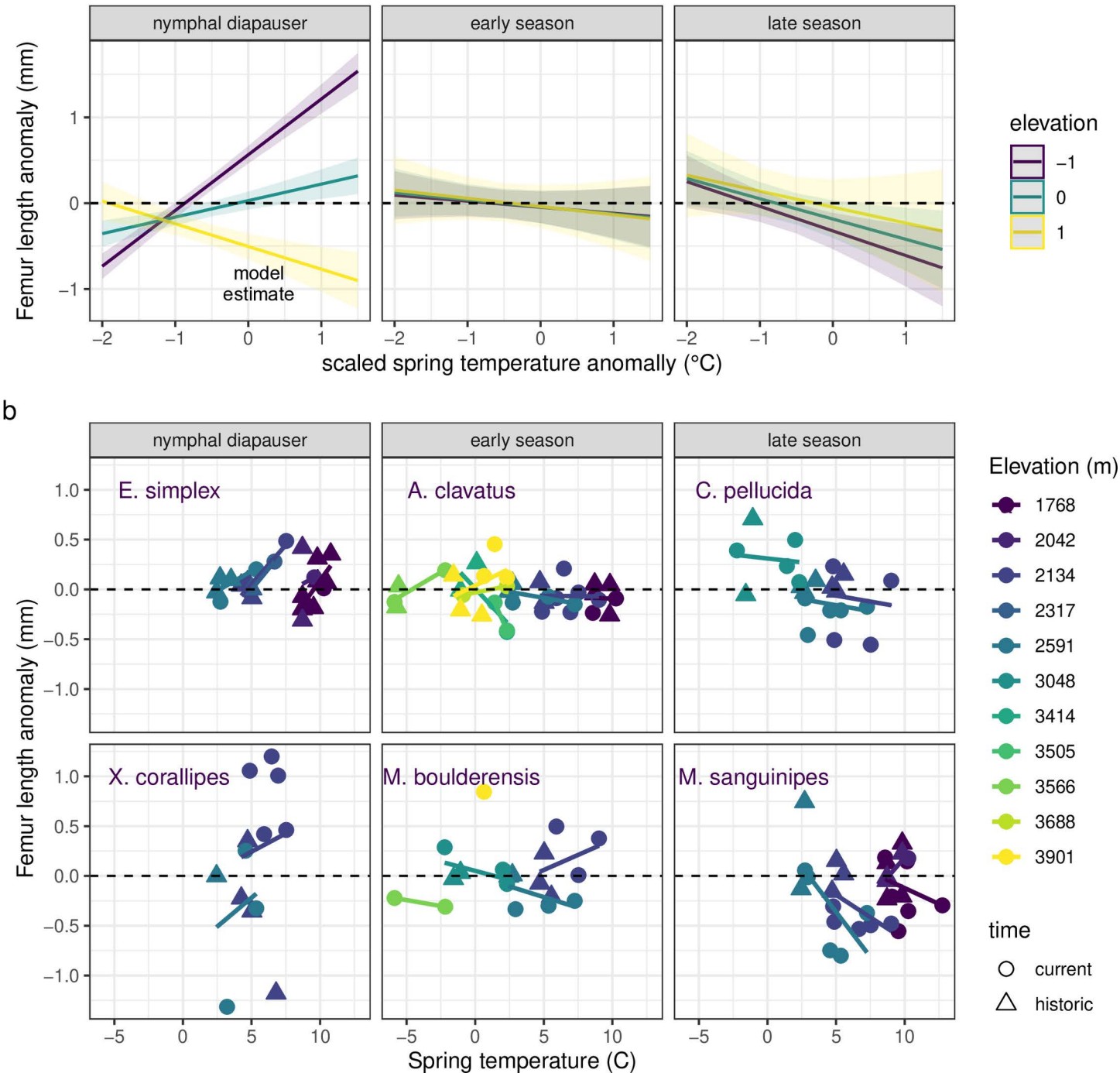

**Fig 5. Seasonal timing, elevation, and the growing season temperature anomalies interact as determinants of changes in femur length from historic means (accounting for species, elevation, and sex).** Nymphal diapausing species have tended to increase size at lower elevations and decrease size at higher elevations in response to warmer seasonal temperatures. Early seasonal species exhibit mixed responses to seasonal temperatures. Converse to the response of nymphal diapausers, late season species have tended to decrease size in response to warmer temperatures, particularly at low elevations. The trends are depicted as **(a)** model estimates ± SE and **(b)** data means ± SE per year and site for species arranged by seasonal timing from the earliest season, nymphal diapausing species to late season species (left to right and top to bottom). Since we did not sample nymphal diapausers at high elevations, we have labeled the high elevation model estimate. Data are plotted as a function of growing season temperatures rather than anomalies to facilitate visualization. The data and code needed to generate this figure can be found in Dryad (https://doi.org/10.5061/dryad.wwpzgmst6).

We attempted to make the temperature predictors more comparable across species by examining the mean temperatures over the 4 weeks before the collection date of each specimen. We term these developmental temperatures, while recognizing that the temperatures might also influence activity and energetics after reaching adulthood. Analogous to the other temperature metrics, we find that early season species exhibit mixed responses and late season species decline in size in response to warmer developmental temperatures (Fig D and Table B in S1 Text). However, we find that nymphal diapausing species consistently increase size in response to warmer developmental temperatures.

### Body size and phenology

Body size is smaller for specimens collected later than average for each species, elevation, and sex across the baseline period (Fig 6). The size response does not vary significantly with elevation or seasonal timing (Table C in S1 Text). For a subset of specimens, we examine associated phenological survey data. We find that earlier phenology corresponds to nymphal diapausers that get bigger at low elevations, but smaller at high elevations. The converse response across elevations is observed for late season species (Fig E in S1 Text).

## Discussion

Natural history collections are an essential resource for investigating phenotypic responses to climate change but often limited by low spatial and temporal replication and collection biases [3,28,29]. We overcame these limitations by leveraging collections resulting from a systematic resurvey project, wherein standardized sampling methods were used on multiple species at specific sites along an elevation gradient in two distinct time periods, bracketing more than 50 years of anthropogenic climate change, where we also have observational and experimental data. Using these highly standardized natural history collections, we found that life history and environmental differences can account for variable relationships between temperature and body size [4]. We found that size responses to warming across elevation were relatively consistent among species with similar seasonal timing, potentially due to responses to thermal opportunity and stress (Fig 1).

Grasshoppers decrease in body size with increasing elevation (Fig 3). This general pattern suggests that climate warming would lead to larger sizes through time. Recent climate warming is steepening elevation clines in body size for early-season grasshopper species and shallowing the clines for later season species via shifts in body size at low elevation (Fig 4). This supports the hypothesis that early season species may be able to better capitalize on warm, early season conditions at low elevation sites with less snow and more early season plant growth [17]. Size increases at low elevation for early season species were consistent with the observed reversed geographic size cline ("hotter is bigger"—larger size in low elevation, warm conditions). In contrast, warming may intensify acquisition or allocation trade-offs for late season species at low elevation sites, for example, through resource limitation due to browning of forage, or energy allocation shifts away from reproduction towards somatic maintenance and repair [21].

Body size changes were most pronounced at low elevations (Fig 4), despite a higher degree of temperature change at high elevations. This differs from predictions that environments with higher exposure to temperature change would show the greatest phenotypic response. In contrast to the observed changes in the slopes of elevation clines, latitudinal clines maintained their slope but shifted in intercept with warming in eight species of passerine birds [41], suggesting that responses may be less variable in endotherms. Additional data on shifts in body size clines in a range of species would be desirable to strengthen these conclusions.

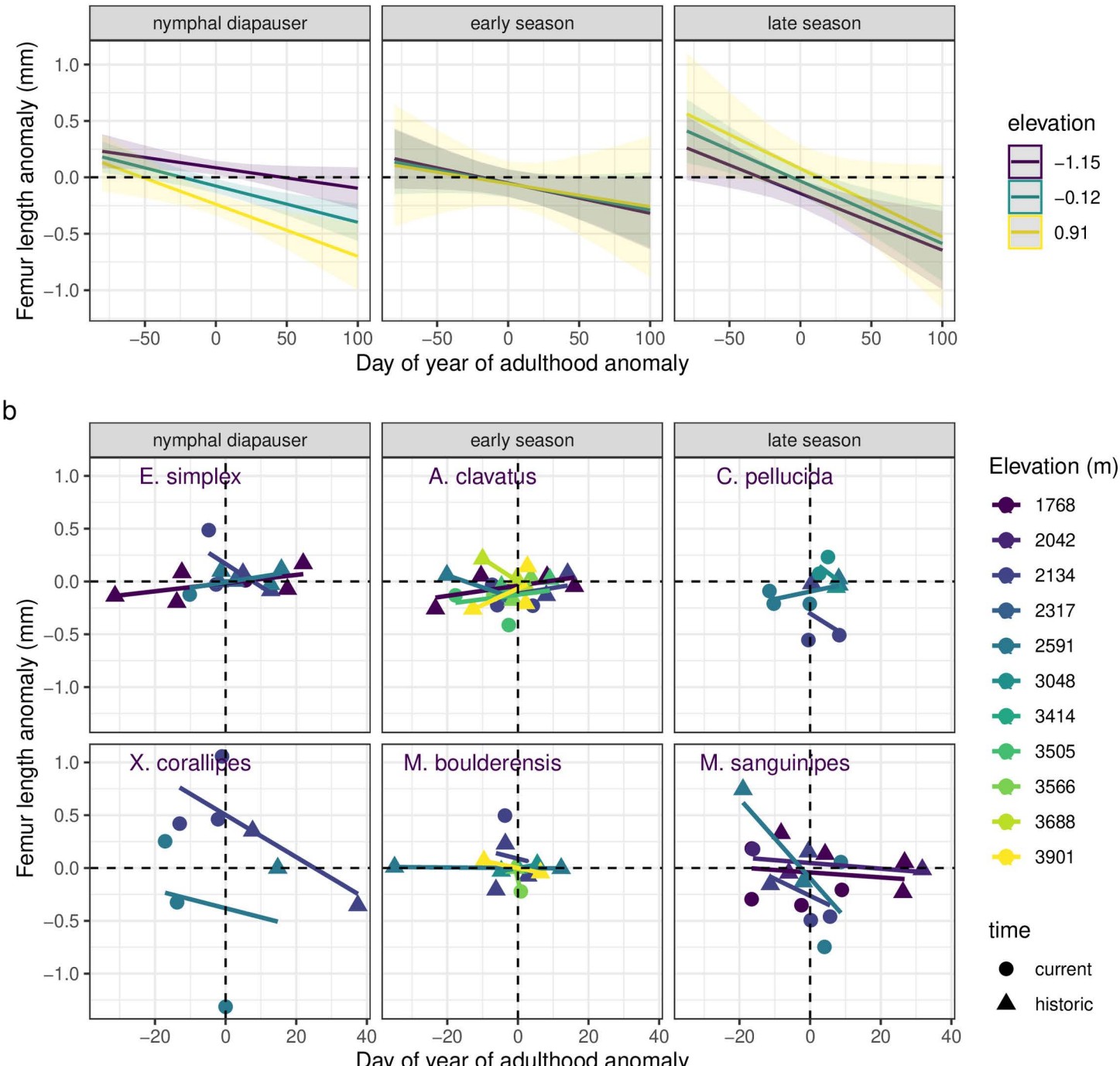

**Fig 6. Femur length anomalies tend to decrease with later phenology (greater day of year of adulthood anomaly) regardless of elevation or species' seasonal timing.** The trends are depicted as (**a**) model estimates ± SE and (**b**) data means ± SE per year and site for species arranged by seasonal timing from the earliest season, nymphal diapausing species to late season species (left to right and top to bottom). Here, phenology is based on the dates of collected museum specimens and we omitted data points representing less than four specimens. See Fig E in S1 Text for the plot assessing phenology based on field surveys. The data and code needed to generate this figure can be found in Dryad (https://doi.org/10.5061/dryad.wwpzgmst6).

The body size changes that we observed through time are most likely attributable to phenotypic plasticity rather than rapid evolutionary adaptation. By leveraging data on interannual temperature variability, we investigated the direction of temperature-size responses for different species and time periods (Fig 5). Growing season temperatures account for the size differences between time periods, suggesting that size differences stem primarily from short-term growth and development responses to temperature rather than longer-term evolutionary changes (Table 1). Similarly, strong temperature responses are evident when examining temperatures across development (Table B in S1 Text). We do; however, detect a persistent four-way interaction between time period, previous summer temperatures, elevation, and seasonal timing that could indicate evolutionary responses (Table A in S1 Text).

Laboratory data on the temperature and photoperiod sensitivity of development and growth supports the potential for differences in developmental plasticity to account for size responses that depend on elevation and species' seasonal timing [42]. The resumption of development for the nymphal diapausing species in our dataset is thought to depend on the interaction of temperature and photoperiod [43]. Thus, at low elevations, where limited snow may allow early plant growth in warm seasons, nymphal diapausers may be able to gain more energy and grow larger before photoperiod cues the completion of development. Temperature and photoperiod also interact to determine development rates for the early season *M. boulderensis*, with high elevation populations exhibiting greater developmental plasticity that can either prolong or accelerate development depending on temperature and photoperiod conditions [39], which may account for the variable size responses of *M. boulderensis*. Higher elevation populations of the late season *M. sanguinipes* tend to develop slower in cool conditions, but faster in warm conditions than lower elevation populations [38]. The rearing experiments bolster our finding here that accelerated development does not necessarily decrease mass [38,39], consistent with grasshoppers capitalizing on permissive conditions.

The differing relationships between developmental timing and body size across elevations and species' seasonal timing correspond to warming posing either an opportunity or stress (Fig 1). Warming can also alter food availability, activity durations, and physiological rates including metabolism and digestion in natural populations, which can lead to variable body size responses [44]. For late season species, conditions enabling earlier phenology may shift from an opportunity to a stress with decreasing elevation. For nymphal diapausing species, the opportunity associated with warming may occur primarily at low elevations with earlier plant growth. Our findings suggest the importance of considering the temperature sensitivity of development and growth to anticipate body size and phenological shifts [45]. Such investigations would benefit from considering differences in temperature sensitivity across life stages [46,47].

Phenological and abundance data from the study sites provide additional context for body size changes. Weekly phenological surveys of our focal species at a subset of our focal sites in 1958–1960 and 2006–2015 documented more pronounced phenological advancements for early developmental stages, for early season species, and at high elevations [36]. These findings are consistent with early season species being better able to capitalize on permissive, warm conditions. Abundance distributions broaden and shift earlier in warm years, increasing phenological overlap between species with different seasonal timing [37]. Greater phenological overlap tends to increase the abundance of early season species, but decrease the abundance of later season species, potentially due to resource competition [37]. Such a dynamic may contribute to our finding that nymphal diapausing species are more likely than later season species to enlarge with warming. The early season is associated with increased thermal opportunity and later season associated with thermal stress to a sufficient extent in our system (Fig 1) that grasshoppers are consistently able to achieve both earlier phenology and larger

size when conditions are permissive. Phenology integrates organismal responses to important environmental drivers and is a valuable resource for anticipating climate change responses when available.

Although the demographic consequences of phenological shifts can be difficult to parse [48], body size provides one important linkage [16]. Clutch mass increases with body size for *M. boulderensis*, *C. pellucida*, and *M. sanguinipes* but not *A. clavatus* (Figs Fand G in S1 Text) [35]. However, earlier development at smaller size could enable the production of more clutches of eggs. Some reproductive metrics such as egg mass and the proportion of functional ovarioles vary with the interaction of body size and elevation (Figs F and G in S1 Text), which alters the implications of the differential size responses across elevation. Further assessments of reproductive output would help interpret and anticipate size responses to climate change.

The body size responses we uncovered, which vary based on elevation and seasonal timing, highlight the benefit of moving beyond universal expectations for responses to warming to better account for the biology and physiology of organisms [49]. A satisfying general theory relating body size to temperature must incorporate causal mechanisms underlying responses that apply to all taxa, and this work illustrates the critical importance of seasonal timing to ectotherms. Shifts in grasshopper body size were consistent with plastic responses to capitalize on warm conditions in the cool, early season and responses to thermal stress in the warm, late season. Despite greater warming at high elevations [40], biological differences have resulted in more pronounced size responses at low elevations. Evaluating how thermal opportunity and stress act along elevation gradients for species with different seasonal timing can inform projections of climate change responses [17].

## Materials and methods

### Specimens and measurements

Grasshoppers were field collected, mostly by sweep netting as part of weekly surveys, in montane or subalpine sites primarily along the 40th N parallel in Boulder County, Colorado. Historical collections were led by Gordon Alexander, with sampling concentrated in 1958–1960. We conducted resurveys with sampling concentrated in 2006–2015. The specimens used to measure body size are available in the University of Colorado Museum of Natural History. Approval for resurvey field research and specimen collection was provided by the US Forest Service, Boulder County Parks and Open Space, City of Boulder Open Space and Mountain Parks Department, and University of Colorado Mountain Research Station.

Body size was measured as femur length, which is the body size metric that can best be compared between museum and recent specimens. Femur length is widely used to indicate body size in grasshoppers and other insects, due to the ease of measurement and its high correlation with body mass [35,50,51]. We measured femurs to the nearest one hundred of a millimeter using digital calipers. Femur length was estimated as the average end-to-end length of both femurs, which were measured twice for most of the specimens. However, a minority of specimens were measured to a lesser resolution, had only one femur, or were measured only once. Historic specimens were dried whereas modern specimens were measured after being stored frozen or were occasionally live when part of other experiments. To confirm that femur length was not altered by drying, we measured some modern specimens fresh and again following being dried.

We focused on species that were well represented in the collection across time periods and elevations. The focal species exhibit additional life history and functional differences. The nymphal diapausers (*E. simplex* and *X. corallipes*), along with *A. clavatus* and *C. pellucida*, primarily feed on grasses and sedges, while the other species are generalists, consuming both

grasses and forbs. The species differ in dispersal ability (due to wing length differences) and lower rates of gene flow result in greater genetic differentiation and more potential for local adaptation [52]. *M. boulderensis* and *A. clavatus* have short wings and are least dispersive. Females of *E. simplex* and *X. corallipes* have longer wings, but tend to be poor fliers.

Our analysis focused on ten sites at the following elevations: 1,768, 2,042, 2,134, 2,317, 2,591, 3,048, 3,414, 3,505, 3,566, and 3,901 m. These sites were chosen because they offered sufficiently large historic and recent body size samples for comparison. The sites are all grassy meadows, with similar plant communities, but somewhat denser vegetation at the mid-elevation sites.

## Climate data

Climate data were obtained for each specimen based on its site and year of sampling. We accessed daily mean temperature data for five weather stations corresponding to collection locations (A1: 2,195 m, 40.01N, 105.37W; B1: 2,591 m, 40.02N, 105.43W; C1: 3,048 m, 40.03N, 105.55W; D1: 3,739 m, 40.06N, −105.62W; Boulder: 1,671 m, 39.99N, 105.27W). The Boulder data were accessed from the NOAA Physical Sciences Laboratory (https://psl.noaa.gov/boulder/data/). Data for other sites were accessed from the Niwot Ridge Long-Term Ecological Research program (LTER, https://nwt.lternet.edu/). Data from 1953 to 2008 were accessed from McGuire and colleagues [40] and included some regression-based interpolation between weather stations to fill data gaps. Data from 2008 to 2015 were accessed from Buckley and colleagues [37] using the same source data interpolation approach [40]. We extended the weather record through 2022 using data from NOAA and the Niwot Ridge LTER site. Following the weather data assembly, data were missing for A1 and B1 from 1970 to 1986. We interpolated this weather data, again using the interpolation approach of McGuire and colleagues [40]. For sites without weather data, we used the available climate data most similar in elevation.

We aggregated daily maximum, minimum, and mean temperature data into averages across days of the growing season (March through August, doy 60–243) and previous summer (June through August, doy 152–243) temperatures. We used temperatures the previous summer since they influence maternal conditions and the development of nymphal diapausing species, which overwinter in a late juvenile stage. We did not examine winter temperatures since most sites have abundant winter snow cover that covers and thermally buffers eggs or nymphs from temperature fluctuations. We examined, but ultimately did not incorporate additional climate metrics including growing degree days available for development and snow depth and timing based on model performance.

## Phenology

We examined how phenology influences body size using the collection date of specimens as well as field surveys. Data from weekly surveys were analyzed for both a historic (1958–1960) and recent (2006–2016) period [36,53]. We quantified phenology as the doy when a spline fit to developmental data indicated that the average developmental stage of the sample population was 5.5 (development index ranging from 1: all first instars to 6: all adults; detailed methods in [36]). Conclusions were similar when phenology was quantified based on the accumulation of heat units available for development [36,53].

## Analyses

Analyses primarily used LMEs models and ANOVA in R (lmer function from Lme4 library). Numeric predictors were centered and scaled (scale function). Models included year nested within species as a random effect. Species was included as a random effect since we are

focused on life history differences between species, but models that included species as a fixed effect performed similarly. We included elevation as a numeric variable since some sites were clustered in elevation and we focused on ten sites spanning elevation. However, we tested that conclusions were similar when treating elevation as an ordered factor. We accounted for seasonal timing using an ordered factor based on species' average phenology across sites [36]. Preliminary models indicated that size responses were similar across sexes, so we omitted sex from our models of climate change responses. We checked for normality of the response variables and subsequently assumed a normal distribution.

We asked sequentially how the body size anomaly varied across time periods, with temperature anomalies, and with both time periods and temperature anomalies. We also accounted for elevation and seasonal timing. Since we expected (biologically) and detected high-order interactions, we included all interactions [Femur anomaly ~ [time period and/or temperature anomaly] * elevation * species timing + (1 | Year:Species)]. We subsequently analyzed whether the body size anomaly varied with phenology, also accounting for elevation and seasonal timing [Femur anomaly ~ doy anomaly * elevation * species timing + (1 | Year:Species)].

## Supporting information

**S1 Text.   Table A. ANOVA results examining how the body size anomaly varies over time (historic or current), with the summer temperature anomaly (Tsum), and with time and the summer temperature anomaly**. We additionally account for elevation (elev) and seasonal timing (SpTiming: nymphal diapauser, early season, late season). We report model Akaike and Bayesian information criterion (AIC and BIC, respectively), where lower values indicate better model performance. **Table B. ANOVA results examining how the body size anomaly varies over time (historic or current), with the developmental temperature anomaly (Tdev), and with time and the developmental temperature anomaly**. We additionally account for elevation (elev) and seasonal timing (SpTiming: nymphal diapauser, early season, late season). We report model Akaike and Bayesian information criterion (AIC and BIC, respectively), where lower values indicate better model performance. **Table C. ANOVA examining changes in the body size anomaly in response to phenology anomaly, elevation (elev), sex, and species. Fig A. Body size (femur length, mm) declines with elevation, females are substantially larger than males, and the body size clines vary with seasonal timing.** We depict both violin plots and population means ± SE by sex and elevation. The species are arranged by seasonal timing from the earliest season, nymphal diapausing species to late season species (left to right and top to bottom). Body size changes between the historic and current time period differ based on seasonal timing and elevation. The slope of the relationship with elevation during the historic period (accounting sex as an intercept) generally declines with later seasonal timing (slope + SE = *Eritettix* simplex: −0.40 ± 0.07, *Xanthippus* corallipes: −0.45 ± 0.22, *Aeropedellus* clavatus: −0.48 ± 0.05, *Melanoplus* boulderensis: −0.35 ± 0.11, *Camnula* pellucida: −0.29 ± 0.13, *Melanoplus* sanguinipes: −0.11 ± 0.22). The data and code needed to generate this figure can be found in Dryad (https://doi.org/10.5061/dryad.wwpzgmst6). **Fig B. Summer temperatures have increased over time.** (a) We depict means of daily mean temperature along with linear regression trends ± SE. (b) Depicting the temperatures as anomalies highlights that temperatures have tended to increase more over recent decades at higher elevations (year: $F[1,343] = 67.01$, $P < 10^{-14}$; elevation: $F[1,343] = 15.46$, $P = 0.001$; year * elevation: $F[1,343] = 14.80$, $P = 0.001$). Hollow dots indicate years where data from other sites were used to fill missing data. The data and code needed to generate this figure can be found in Dryad (https://doi.org/10.5061/dryad.wwpzgmst6). **Fig C. Seasonal timing, elevation, and previous**

**summer temperature anomalies interact as determinants of changes in femur length from historic means (accounting for species, elevation, and sex).** Nymphal diapausing species have tended to increase size at lower elevations and decrease size at higher elevations in response to warmer spring temperatures. Early seasonal species exhibit mixed responses to spring temperatures. Late season species have tended to decrease size, more strongly at lower elevations, in response to warmer temperatures the previous summer. The trends are depicted as **(a)** model estimates ± SE and **(b)** data means ± SE per year and site for species arranged by seasonal timing from the earliest season, nymphal diapausing species to late season species (left to right and top to bottom). Data are plotted as a function of spring temperatures rather than anomalies to facilitate visualization. The data and code needed to generate this figure can be found in Dryad (https://doi.org/10.5061/dryad.wwpzgmst6). **Fig D. Seasonal timing, elevation, and developmental temperature interact to determine changes in femur length in grasshoppers.** Nymphal diapausing species have a positive temperature-size response ("hotter is bigger"), with all populations, but especially high elevation increasing in size in response to warmer developmental temperatures. Mid-season species do not show consistent responses to developmental temperatures. Late season species have a negative temperature-size response ("hotter is smaller"), tended to decrease size, more strongly at lower elevations, in response to warmer temperatures the previous summer. The trends are depicted as **(a)** model estimates ± SE and **(b)** data means ± SE per year and site for species arranged by seasonal timing from the earliest season, nymphal diapausing species to late season species (left to right and top to bottom). Femur length anomalies are mean femur lengths in the modern samples (dates) expressed relative to historic means (dates), accounting for species, elevation, and sex. Developmental temperatures are mean temperatures over the 30 days before the collection data of each specimen. Data are plotted as a function of developmental temperatures rather than anomalies to facilitate visualization, but were analyzed as anomalies. The data and code needed to generate this figure can be found in Dryad (https://doi.org/10.5061/dryad.wwpzgmst6). **Fig E. Femur length anomalies tend to decrease with later phenology (greater day of year of adulthood anomaly) when phenology is based on field surveys.** The trends are depicted as **(a)** model estimates ± SE and **(b)** data means ± SE per year and site for species arranged by seasonal timing from the earliest season, nymphal diapausing species to late season species (left to right and top to bottom). The data and code needed to generate this figure can be found in Dryad (https://doi.org/10.5061/dryad.wwpzgmst6). **Fig F. Reproductive metrics including (from top to bottom) egg mass, clutch mass, the number of ovarioles, and the proportion of functional ovarioles vary with femur length and elevation (color).** Data points correspond to individuals and we depict linear regression trends and standard errors. Data are from Levy and Nufio [35]. The data and code needed to generate this figure can be found in Dryad (https://doi.org/10.5061/dryad.wwpzgmst6). **Fig G. Model estimates ± SE indicate that reproductive metrics including (from top to bottom) egg mass, clutch mass, the number of ovarioles, and the proportion of functional ovarioles vary with femur length and elevation (color).** Data are from Levy and Nufio [35]. The data and code needed to generate this figure can be found in Dryad (https://doi.org/10.5061/dryad.wwpzgmst6). (PDF)

## Acknowledgments

We thank the Niwot Ridge LTER program for access to weather data and the University of Colorado Museum of Natural History for access to the specimens. We thank Maria Cruz-Lopez, Jeff McClenahan, Ali Moore, Virginia Scott, Johannna Zeh, Rick Levey and other students and volunteers for helping to collect and measure grasshoppers. We thank members of our research groups for input or comments on the manuscript.

## Author contributions

**Conceptualization:** César R. Nufio, Monica M. Sheffer, Lauren B. Buckley.

**Data curation:** César R. Nufio, Lauren B. Buckley.

**Formal analysis:** César R. Nufio, Monica M. Sheffer, Julia M. Smith, Lauren B. Buckley.

**Funding acquisition:** César R. Nufio, Sean D. Schoville, Lauren B. Buckley.

**Investigation:** César R. Nufio, Monica M. Sheffer, Julia M. Smith, Michael T. Troutman, Sean D. Schoville, Lauren B. Buckley.

**Methodology:** César R. Nufio, Lauren B. Buckley.

**Project administration:** César R. Nufio, Sean D. Schoville, Caroline M. Williams, Lauren B. Buckley.

**Resources:** César R. Nufio, Lauren B. Buckley.

**Software:** Lauren B. Buckley.

**Supervision:** César R. Nufio, Sean D. Schoville, Caroline M. Williams, Lauren B. Buckley.

**Visualization:** Monica M. Sheffer, Lauren B. Buckley.

**Writing – original draft:** César R. Nufio, Monica M. Sheffer, Caroline M. Williams, Lauren B. Buckley.

**Writing – review & editing:** César R. Nufio, Monica M. Sheffer, Julia M. Smith, Michael T. Troutman, Simran J. Bawa, Ebony D. Taylor, Sean D. Schoville, Caroline M. Williams, Lauren B. Buckley.

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
