## [Editor Report · Decision Letter 0]

12 Aug 2024

Dear Lauren,

Thank you for submitting your manuscript entitled "Insect size responses to climate changes vary across elevations according to seasonal timing" for consideration as a Research Article by PLOS Biology.

Your manuscript has now been evaluated by the PLOS Biology editorial staff, as well as by an academic editor with relevant expertise, and I'm writing to let you know that we would like to send your submission out for external peer review.

Once your full submission is complete, your paper will undergo a series of checks in preparation for peer review. After your manuscript has passed the checks it will be sent out for review. To provide the metadata for your submission, please Login to Editorial Manager (https://www.editorialmanager.com/pbiology ) within two working days, i.e. by Aug 14 2024 11:59PM.

Kind regards,

Roli

Roland Roberts, PhD

Senior Editor

PLOS Biology

rroberts@plos.org

---

## [Decision Letter · Decision Letter 1]

16 Oct 2024

Dear Lauren,

Thank you for your patience while your manuscript "Insect size responses to climate changes vary across elevations according to seasonal timing" went through peer-review at PLOS Biology. Your manuscript has now been evaluated by the PLOS Biology editors, an Academic Editor with relevant expertise, and by three independent reviewers.

You'll see that reviewer #1 is very positive about the study, but does (in his enthusiasm) suggest a few more analyses that he says are nice but optional. He suggests that you improve Fig S4 and move it to the main paper, and has a load of textual and presentational suggestions, some of which may involve extra analyses. Reviewer #2 is also positive, and simply wants you to connect it better to the existing literature. Reviewer #3 is positive, but thinks that the motivation is weak, and (like rev #1) suggests promoting Fig S4 to the main paper. S/he has a long list of other textual and presentational requests, and one or two might need a minor analysis.

In light of the reviews, which you will find at the end of this email, we are pleased to offer you the opportunity to address the comments from the reviewers in a revision that we anticipate should not take you very long. We will then assess your revised manuscript and your response to the reviewers' comments with our Academic Editor aiming to avoid further rounds of peer-review, although might need to consult with the reviewers, depending on the nature of the revisions.

**IMPORTANT - SUBMITTING YOUR REVISION**

*Resubmission Checklist*

*Published Peer Review*

*PLOS Data Policy*

Please note that as a condition of publication PLOS' data policy (http://journals.plos.org/plosbiology/s/data-availability ) requires that you make available all data used to draw the conclusions arrived at in your manuscript. If you have not already done so, you must include any data used in your manuscript either in appropriate repositories, within the body of the manuscript, or as supporting information (N.B. this includes any numerical values that were used to generate graphs, histograms etc.). For an example see here: http://www.plosbiology.org/article/info%3Adoi%2F10.1371%2Fjournal.pbio.1001908#s5

*Blot and Gel Data Policy*

Sincerely,

Roli

Roland Roberts, PhD

Senior Editor

PLOS Biology

rroberts@plos.org

REVIEWERS' COMMENTS:

Reviewer #1:

[identifies himself as Wilco C.E.P. Verberk]

General comments:

I very much enjoyed reading this study by César Nufio and co-workers in which they focus on body size responses across elevational clines and multidecadal changes in climate. With 6 species, two time periods and many different elevations there is a lot to unpack and the strength of this study lies in the clarity with which things are presented and explained as well as the unique dataset. I made some suggestions below, which will hopefully help the researchers to further improve or clarify their manuscript. In my enthusiasm, I also suggested some additional analyses; I realise the supplementary materials is already packed with analysis and the author will be in the best position to decide if those analysis are useful or only lead to rabbit holes that are either already explored or will detract from the main message of this manuscript.

Signed, Wilco Verberk

Detailed comments:

Line 51: this statement about small-bodied organisms likely pertains especially to terrestrial small-bodied organisms. Note that for aquatic animals there are patterns between body size and heat tolerance limits, but these are generally negative, i.e. small bodies organisms have higher peak body temperatures.

Line 54-59: Note also figure 5 in ref 7 where a link between voltinism and size responses is made. Voltinisms may be particularly important in terrestrial environments where end-of-season-constraints are harsh (e.g. drought and frost), which are attenuated at least somewhat in an aquatic environment.

Line 69: In this light, it may be better to replace 'insects' in the title by 'grasshoppers'?

Line 81: I found fig S4 potentially worthwhile to include as a conceptual figure in the introduction, but somewhat confusing in its present form: in the left panel, I see all different hypotheses, but some very specific ones were made (as far as I could tell, the intro suggests the ascending line of low elevation shrinkage and high elevation enlarging). The y-axis is basically a temperature axis as it compares historic and current sizes and the main difference is temperature. In that sense the right panel should not really have a thermal cline as the comparison is already between historic and present which incorporates a temperature comparison. So I don't follow what the different lines do. Note that this panel would make a lot more sense if it just had size on the y-axis rather than anomaly. Also, since the paper to this point has been mostly about temperature (not altitude or historical warming), it may be clearer to lead with that panel.

Line 90: Although the introduction lays out the framework and an overarching hypothesis, the data are quite complex and breaking it down into subquestions each addressed by a different analysis and paragraph could help. For example, this paragraph asks about the elevational cline and whether this is different depending on a species' seasonal timing. This is important to establish but does not yet deal with the comparison between historical or current data. In the next paragraph this follow up question could then be asked. Note that from Table 1, there does not seem to be a strong effect of Time, which agrees with me eyeballing figure 2. 

Line 92: I can see why the size cline is referred to as a hotter is better pattern, but the high elevation populations presumably have less time and the greater time constraints likely select for faster development or a greater thermal sensitivity of development rates (as I see has been observed for M. boulderensis in line 214). So this pattern could have also arisen by effects other than temperature. Note that with all the temperature data, it is possible to convert the elevational points into a temperature cline and then see if decreasing body size with increasing elevation can be explained by increasing temperatures at low elevation. By including interactions between this temperature gradient and both altitude and time, one could even test to what extent temperature size-responses are modulated by either altitude or evolution.

Line 92: To drive home the point that the slopes of the elevational relationships are shallower with later seasonal timing, I would plot all panels with the same range for the x-axis and y-axis and express the slopes as a percentage of 'shrinkage' to account for size differences across species.

Line 99-105: This is basically the stats to underpin the previous paragraph? So maybe just incorporate there? Also, were two analyses performed (one on historic data, one on current data? Why not include historic and current as a factor to provide different intercepts?

Line 102: How do these F-values relate to those quoted in table 1?

Line 104: Given the significant 3-way interaction that includes sex, I wonder why sex does not feature in much of the rest of the manuscript? Many of the reasoning (e.g. capitalizing on permissive conditions for growth) will be more or less important depending on the sex of the animal. Note that this interaction also seems to run against line 339

Line 109: This refers to the time:elev:SpTiming interaction in table 1 (P=0.02)? Not that this effect is not included in the best model and not significant in a model that also includes temperature (P = 0.76).

Line 113: I guess figure 3 is based on the first model in Table 1? I would suggest to express the femur length anomaly as a percentage to prevent larger species to have greater anomalies just because they are larger, rather than being more plastic in size.

Line 118: In figure 4, it seems that femur anomalies are all over the place for nymphal diapausers, until I realised that both species are both inhabiting the lower altitudus. Instead of showing the full range of the (scaled) elevation in these plots, maybe only show model preductions for the upper and lower elevational range for the species involved (I very much appreciate the constancy of colors to denote elevation)?

Line 145 So, if you were to plot absolute doy vs absolute size for each of the two time periods one would see a negative relationship (i.e. animals taking longer to develop will reach a smaller size at adulthood)? This is strange in the context of the TSR where one frequently assumes there is a case of temperature shifting the trade-off towards maturing early at a small size rather than later at a large size.

Line 175 In this sense it could be interesting to see if the size-doy relationship for these late season grasshoppers is different in anomalously hot years or not.

Line 179: maybe I missed this, but where is this in the results? I think this is a bit tangential as the paper focusses on seasonal timing of species as an important driver for size anomalies. Also, since nymphal diapausing species (with steep slopes) are low elevation, I wonder to what extent one can link body size changes unambiguously to elevation. If you need to cut something from the ms, I think this paragraph is a good candidate unless I missed something.

Line 190 Something which is further corroborated by the strong effect of temperature in the 4 weeks preceeding adult measurements?

Line 216 Allow me to highlight a study in butterflies where growth and development were also modulated by an interaction between temperature and photoperiod, although the size response was much buffered as growth and development were similarly affected (https://doi.org/10.1016/j.cris.2022.100034 )

Line 220: Instead of referring to fig 5, this could refer to (a modified, clearer version of) Fig S4?

Line 230: A problem with multi-species comparison, is that species will differ in more respects than that which one is interested at. A possible solution could be here to focus on differences between males and females. Do males and females differ in their trade-off between growth (ie size) and development (i.e. time or doy) and how is this influenced by cues of temperature and photoperiod. For males, it is beneficial to not just grow as large as possible, filling up the whole period of permissible growth season, but rather to also be sufficiently early to maximise mating opportunities. I wonder if the 3-way interaction between elevation, sex and seasonal timing (line 104) can be leveraged to study how prioritizing size and age is a function of temperature and time available (i.e. photoperiod).

Reviewer #2:

In the MS "Insect size responses to climate changes vary across elevations according to seasonal timing" by Nufio et al. the authors explore the effect of warming on insect development.

The study reports an increase of growth season temperatures along an elevational gradient using temperature records for the last 60 y. The study compares length of femurs as an estimate of size for six species of grasshoppers. The life histories of these species show different seasonal timing that interact with changing temperatures along the elevational gradient. The study shows that size responses are complex and depends on the timing at which species start their development. Early species show an increase of size, but late species show a decrease of size.

This study takes advantage of a unique historical dataset to determine the complex effects of increasing temperatures on insect morphology, and maybe population dynamics. My only concern is that this study needs to connect the results with the broader context, and general theory.

My suggestion to the authors is to include in the introduction some of the theoretical expectations of ecogeographic rules.

For example, the authors should discuss how their findings connect to previously proposed patterns such as 'James' rule', which focuses on the relative effect of temperature and body size:

James, F. C. (1970). Geographic size variation in birds and its relationship to climate. Ecology, 51, 365-390. https://doi.org/10.2307/1935374

Or the 'temperature-size rule', which states that body size tends to decrease with increasing temperature.

Atkinson, D. (1994). Temperature and organism size: A biological law for ectotherms? Advances in Ecological Research, 25, 1-58.

The manuscript will also benefit to discuss if ectotherms should be considered under Bergmann's rule.

Blackburn, T. M., Gaston, K. J., & Loder, N. (1999). Geographic gradients in body size: A clarification of Bergmann's rule. Diversity and Distributions, 5, 165-174.

Daufresne, M., Lengfellner, K., & Sommer, U. (2009). Global warming benefits the small in aquatic ecosystems. Proceedings of the National Academy of Sciences of the United States of America, 106, 12788-12793.

In the discussion, the authors may include the potential effects of oxygen partial pressure and air density.

Dillon, M. E., Frazier, M. R., & Dudley, R. (2006). Into thin air: Physiology and evolution of alpine insects. Integrative and Comparative Biology,

Callier, V., & Nijhout, H. F. (2011). Control of body size by oxygen supply reveals size-dependent and size-independent mechanisms of molting and metamorphosis. Proceedings of the National Academy of Sciences of the United States of America, 108, 14664-14669

Also, the authors may discuss the potential phylogenetic effects on their results.

After these minor changes, I think the study is an excellent contribution to understand how ectotherm size is affected by increasing temperatures.

Reviewer #3:

Paper Summary

In this paper, the authors used natural history collections and modern field observations to investigate how the body sizes of grasshoppers have changed over 60 years of recent climate change. The study system was an assemblage of six grasshopper species living along a 2000-m elevation gradient in the Rocky Mountains. The authors compared the body sizes of grasshoppers collected from 1958-1960 to the body sizes of modern samples collected from 2006-2015. By fitting linear mixed-effect models, the authors analyzed how species traits interact with climate, time, and elevation to influence the body size of grasshoppers. These analyses enabled the authors to evaluate competing hypotheses for how temperature affects body size ("hotter is smaller" vs "hotter is bigger") while documenting changes in body size in response to climate change. The authors found that body size responses varied among species and life histories. As the climate has warmed, early-season species, especially those at low elevations, tended to develop faster and grow larger. In contrast, warming reduced the body sizes of grasshoppers that develop later in the season and live at higher elevations, underscoring the idiosyncratic nature of climate change responses and the complex relationships between temperature and body size.

Strengths

The greatest strength of this paper is its successful integration of multiple approaches. The authors combined historical collections, modern sampling, and quantitative analyses to document climate change responses and test eco-physiological hypotheses. These datasets and analyses are thoroughly described and scientifically sound, and the authors have produced clear figures that effectively communicate the results.

Weaknesses

A weakness of this paper is the stated motivation; the authors assert that declining body size has been proposed as a universal response to warming. In my view, no reasonable biologist would predict a universal response, and relatively recent investigations of this topic have confirmed that there is variation in how body size responds to warming (e.g., Diamond et al. 2010; Kingsolver and Huey 2008; Ashton and Feldman 2003). I believe that the paper would be strengthened by a focus on competing hypotheses for body size responses, rather than evaluating evidence for a "universal response". I especially appreciate the ideas behind the author's conceptual S4 Figure, which outlines two competing hypotheses. Although I am confused by some of the lines in the left panel, I recommend elevating this figure to the main text and framing the study as an investigation of those competing hypotheses.

Paper Assessment

Overall, this paper successfully integrates approaches to investigate the impacts of climate change on grasshoppers. It is a well-written paper and well-conceived project that will be of interest to a broad group of ecologists, physiologists, and entomologists. I hope that my comments can reinforce the clarity and impact of the paper.

Specific comments

Abstract

Overall: This abstract would be more effective if it included more details about the study and results. My main recommendation is to state how exactly body size and seasonal timing have changed for these grasshoppers on average (e.g., % change in femur length). Similarly, consider specifying more about the study design, including the time periods, number of species, elevation range, etc.

L18-20: I am wondering how many biologists would argue today that body size declines are a "universal response to warming". Huey and Kingsolver 2008 (cited by the authors) found that warming reduced the body size of only 80% of the studied organisms, demonstrating 16 years ago that this response is not universal. Consider revising this opening sentence to identify a clearer knowledge gap and clarify the prevalence of body size declines among diverse taxa.

L21-22: It seems like "universal size responses" is being used as a stand-in for predictions from one hypothesis (i.e., temperature-size rule), which is known to not be universal. Is there a more nuanced way to present the evidence of body size responses?

L23-24: Starting at "Size shifts have been…" I think there is a misplaced comma in this sentence, and the sentence is hard to follow. Consider splitting into two sentence or otherwise revising for clarity.

L27: Please clarify whether this hypothesis was formulated a priori or after the results were known. If the former, I recommend introducing this hypothesis earlier in the abstract. If the latter, I recommend making it clear that the authors used the results to formulate the hypothesis.

Introduction

L37-38: Citations 1 and 2 seem to also refute the idea of universal size responses to warming. There is reason to believe that many organisms will shrink, but few biologists, if any, would have argued that this response would be universal… Perhaps this first sentence of the introduction can be deleted to make way for an argument from first principles as to why body size declines (or increases) might be expected.

L46: I believe there should be a dash between temperature and size (temperature-size rule)

L46-48: Consider introducing Bergmann's Rule, which seems relevant to the TSR and the relative support for endotherms vs ectotherms.

L68-74: I enjoyed reading this clear description of why grasshoppers tend to reverse the TSR.

L78-81: This section presents a detailed hypothesis that is partly refuted and partly supported by the subsequent analyses, yet the abstract gives the impression that the authors' findings were entirely "consistent with our hypothesis". Consider fully detailing this hypothesis in the abstract.

Results

L93-98: This information about the species and their seasonal timing might make more sense in the Methods to illustrate how the authors deliberately selected the species to control for other sources of variation (e.g., in voltinism).

L97: The authors call A. clavatus and M. boulderensis mid-season species here, but elsewhere, it seems like "early-season" is used.

L104: The methods (L339) state that sex was omitted from the models, yet sex differences are graphed (Fig. 2) and reported. Please clarify in the methods

L109-112: Add a semi colon in place of the comma on L112 to fix the run-on sentence

L111: In Figure 3A, I am surprised that the model estimates such weak effects of elevation on the body size of mid-season species (M. boulderensis and A. clavatus) because there seem to be strong body size clines for these species in Figure 2. What causes this discrepancy?

L134-136: Consider adding more detail to explain biologically what this 4-way interaction implies. This might make it easier for readers to see how this is evidence for evolutionary change. For example, "The 4-way interaction suggests that grasshoppers at different elevations and with different life histories are responding differently to temperature now than they did historically" or something to that effect.

L136: Is "Extended Data Table S1" different from "Table S1"?

L145-150: The objective(s) of these analyses were not clear to me.

Figure 2: These graphs are not intuitive because they all have different scales on the x and y axes. Consider standardizing the scales wherever possible, although I know that will create other data visualization problems.

Figure 3A: The yellow shading is quite hard to see. Consider using another color with more contrast. Also, double check the paper for use of nymphal diapauser vs early season vs mid season. The authors introduce "mid-season" on L97, but it is not used elsewhere. I happen to like the term mid-season to more easily distinguish from nymphal diapausers.

Discussion

Overall: Discussion would be strengthened by adding references to tables and figures to remind readers of where to find evidence for various claims.

L159-160: The authors' focus on universal responses is misplaced. A universal response would be evidenced by every species always responding in the same direction, but no reasonable biologist would have predicted a universal response in body size. I think the paper and its conclusions would be stronger if the authors framed their study and conclusions as providing evidence to distinguish between competing hypotheses, rather than as providing evidence for or against a "universal response".

L162-164: What statistical evidence supports the claim that "seasonal timing was the most important predictor of response to climate change"? I am specifically wondering how we know that seasonal timing is a more important predictor than say species. Also, this statement that "changes were explained by divergent phenotypic plasticity in species with contrasting life histories" seems too strongly worded. Given the correlative nature of the analyses, how confident are the authors that these changes are explained by phenotypic plasticity? And how confident are the authors that contrasting life histories is the mediating factor? It seems to me that this is best framed as a newly formed hypothesis, not a final conclusion.

L168: Note that life history differences are just one explanation for the different responses to climate warming. There are many other traits that might be better predictors but were not included here.

L170: I recommend referencing Figure 3 at the end of this sentence.

L186-191: The authors also found strong support for a 4-way interaction with temperature, time period, seasonal timing, and elevation, supporting the idea that the body size response to temperature is different today than it was previously. How does this evidence factor into their conclusions about evolutionary change vs phenotypic plasticity?

L205-208: What is meant by "completion" here? Is it when the nymphs arrest development ahead of overwintering? Or is it when nymphs completely develop to adulthood following overwintering?

L220: I do not understand why the authors are referencing Figure 5 here. Perhaps a typo and Figure S4 would be better?

L243: I do not see how the effect of phenology is a universal response when there are positive and negative slopes shown in Fig S8, panel A.

Methods

L284: How were the focal species selected? I assume that the historical collections and modern sites have more species than those studied here?

L291: Replace "spanning" with "at"

L333-334: I would appreciate more justification for why the authors included species as a random effect rather than a fixed. I suspect that it is related to the fact that the authors are primarily aiming to estimate the effects of life history, not species, on femur length, but it would be good to clarify. Similarly, I am curious whether model performance is higher when species is included as a fixed effect instead of life history.

---

## [Editor Report · Decision Letter 2]

3 Dec 2024

Dear Lauren,

Thank you for your patience while we considered your revised manuscript "Insect size responses to climate changes vary across elevations according to seasonal timing" for publication as a Research Article at PLOS Biology. This revised version of your manuscript has been evaluated by the PLOS Biology editors and the Academic Editor.

Based on our Academic Editor's assessment of your revision, we are likely to accept this manuscript for publication, provided you satisfactorily address the following data and other policy-related requests.

a) Please can you modify your Title very slightly by removing the "s" from "changes"? We appreciate that you did assess multiple aspects of climate change, but this will be readily apparent in your Abstract, and the presence of the phrase "climate change" in the Title will, we think, enhance discoverability.

b) You rightly say that no ethical approval was needed, but did you require a field research licence for the sample collection? Please include this information if applicable.

c) Please address my Data Policy requests below; specifically, we need you to supply the numerical values underlying Figs 2AB, 3, 4AB, 5AB, 6AB, S4, S5AB, S6AB, S7AB, S8AB, S9, S10, either as a supplementary data file or as a permanent DOI’d deposition. I note that you already have an associated GitHub deposition, containing a nicely organised set of code that I suspect, along with the data in Dryad, might enable readers to reconstruct the Figs. Please can you confirm this?

d) Because Github depositions can be readily changed or deleted, please make a permanent DOI’d copy (e.g. in Zenodo) and provide this URL (see below). An potentially neater alternative would be to dump a copy of the Github code into your Dryad deposition, so that everything's in the same place...

e) Please cite the location of the data clearly in all relevant main and supplementary Figure legends, e.g. “The data underlying this Figure can be found in S1 Data” or “The data underlying this Figure can be found in https://zenodo.org/records/XXXXXXXX " or “The data and code needed to generate this Figure can be found in https://doi.org/10.5061/dryad.wwpzgmst6 "

We expect to receive your revised manuscript within two weeks.

*Published Peer Review History*

*Press*

Sincerely,

Roli

Roland Roberts, PhD

Senior Editor

rroberts@plos.org

PLOS Biology

DATA POLICY:

You may be aware of the PLOS Data Policy, which requires that all data be made available without restriction: http://journals.plos.org/plosbiology/s/data-availability . For more information, please also see this editorial: http://dx.doi.org/10.1371/journal.pbio.1001797

Regardless of the method selected, please ensure that you provide the individual numerical values that underlie the summary data displayed in the following figure panels as they are essential for readers to assess your analysis and to reproduce it: Figs 2AB, 3, 4AB, 5AB, 6AB, S4, S5AB, S6AB, S7AB, S8AB, S9, S10. NOTE: the numerical data provided should include all replicates AND the way in which the plotted mean and errors were derived (it should not present only the mean/average values).

CODE POLICY

DATA NOT SHOWN?

---

## [Editor Report · Decision Letter 3]

17 Dec 2024

Dear Lauren,

Thank you for the submission of your revised Research Article "Insect size responses to climate change vary across elevations according to seasonal timing" for publication in PLOS Biology. On behalf of my colleagues and the Academic Editor, Andrew Tanentzap, I'm pleased to say that we can in principle accept your manuscript for publication, provided you address any remaining formatting and reporting issues. These will be detailed in an email you should receive within 2-3 business days from our colleagues in the journal operations team; no action is required from you until then. Please note that we will not be able to formally accept your manuscript and schedule it for publication until you have completed any requested changes.

We also ask that you take this opportunity to read our Embargo Policy regarding the discussion, promotion and media coverage of work that is yet to be published by PLOS. As your manuscript is not yet published, it is bound by the conditions of our Embargo Policy. Please be aware that this policy is in place both to ensure that any press coverage of your article is fully substantiated and to provide a direct link between such coverage and the published work. For full details of our Embargo Policy, please visit http://www.plos.org/about/media-inquiries/embargo-policy/ .

Best wishes,

Roli

Senior Editor

PLOS Biology

rroberts@plos.org